# *Nidus vespae* Built by an Invasive Alien Hornet, *Vespa velutina* *nigrithorax*, Inhibits Adipose Tissue Expansion in High-Fat Diet-Induced Obese Mice

**DOI:** 10.3390/biology11071013

**Published:** 2022-07-06

**Authors:** Seul Gi Lee, Dong Se Kim, Jongbeom Chae, Eunbi Lee, Dongyup Hahn, Il-Kwon Kim, Chang-Jun Kim, Moon Bo Choi, Ju-Ock Nam

**Affiliations:** 1Department of Food Science and Biotechnology, Kyungpook National University, Daegu 41566, Korea; lsg100479@naver.com (S.G.L.); aodydirk@naver.com (D.S.K.); chejongbum@naver.com (J.C.); 21eunbi@naver.com (E.L.); 2Department of Immunology, School of Medicine, Keimyung University, Daegu 42601, Korea; 3School of Food Science and Biotechnology, College of Agriculture and Life Sciences, Kyungpook National University, Daegu 41566, Korea; dohahn@knu.ac.kr; 4Department of Integrative Biology, Kyungpook National University, Daegu 41566, Korea; 5Division of Forest Biodiversity, Korea National Arboretum, Pocheon 30106, Korea; ilkwons91@forest.go.kr (I.-K.K.); changjunkim@korea.kr (C.-J.K.); 6Institute of Plant Medicine, Kyungpook National University, Daegu 41566, Korea; 7Research Institute of Tailored Food Technology, Kyungpook National University, Daegu 41566, Korea

**Keywords:** adipogenesis, adipose browning, insulin signaling, *Nidus vespae*, obesity

## Abstract

**Simple Summary:**

*Nidus vespae* (NV) has been used as an ingredient in crude drugs in Korea and China. However, the effects of NV on obesity and its mechanism have not been completely elucidated. Herein, we demonstrated the novel anti-obesity effects of NV in in vivo and in vitro systems. The administration of NV ameliorated adipose expansion and improved adipose browning in white adipose tissue from high-fat diet (HFD)-fed mice. Moreover, treatment with NV suppressed adipocyte differentiation, possibly through inactivation of the insulin signaling pathway in adipocytes.

**Abstract:**

*Nidus vespae*, commonly known as the wasp nest, has antioxidative, anti-inflammatory, antimicrobial, and antitumor properties. However, the anti-obesity effects of *Nidus vespae* extract (NV) have not yet been reported. This study aimed to elucidate the potential anti-obesity effects of NV in vivo and in vitro, using a high-fat diet (HFD)-induced obese mouse model and 3T3-L1 adipocytes, respectively. NV administration to HFD-induced obese mice significantly decreased the mass and plasma lipid content of adipose tissues. Uncoupling protein-1 expression was significantly higher in the inguinal white adipose tissues of NV-treated mice than in those of HFD-fed mice. Furthermore, we found that NV inhibited the differentiation and intracellular lipid accumulation of 3T3-L1 adipocytes by regulating the insulin signaling cascade, including protein kinase B, peroxisome proliferator-activated receptor gamma, CCAAT/enhancer binding protein alpha, and adiponectin. These findings suggest that NV may exhibit therapeutic effects against obesity by suppressing adipose tissue expansion and preadipocyte differentiation, thereby providing critical information for the development of new drugs for disease prevention and treatment. To our knowledge, this study provides the first evidence of the anti-obesity effects of NV.

## 1. Introduction

Obesity is a prominent public health problem and a major risk factor for various diseases. Abnormal adipocyte (fat cell) differentiation is characterized by adipocyte hyperplasia (increased number) and hypertrophy (increased size), ultimately leading to obesity [1]. Adipocytes are classified into the following three types: white, beige, and brown adipocytes [2]. Beige and brown adipocytes dissipate energy as heat through mitochondrial uncoupling protein-1 (UCP1), whereas white adipocytes function primarily to store excess energy in the form of triglycerides (TGs) [2]. From this perspective, adipose browning, the process through which white adipocytes can change their phenotype and function as brown- and beige-like adipocytes, confers beneficial effects for the treatment of metabolic diseases, such as obesity and diabetes [3]. 

Edible insects have been widely used as food or traditional medical materials by humans for a long time. However, owing to the development of large-scale agriculture, food, and pharmaceutical industries, edible insects have been reduced to hate foods in most countries, except some in Africa, Southeast Asia, and Latin America (where foods are used as edible or folk remedies) [4]. However, as edible insects are being re-examined as alternative food and pharmaceutical ingredients, interest in their usefulness is increasing [4]. In fact, edible insects are more eco-friendly, grow faster, show higher feed conversion efficiency, and contain more abundant nutrients, such as vitamins, minerals and natural therapeutic substances, than livestock. Hence, the scale of the edible insect industry is expected to grow in the future [4,5,6,7,8]. Currently, about 2000 species of edible insects, mainly beetles, caterpillars, bees, wasps, grasshoppers, crickets, and flies, are known worldwide [8,9]. Since studies on the extraction and efficacy of various substances from various potential and already licensed edible insects are being conducted, the number of food and pharmaceutical raw materials obtained from insects is expected to increase in the future [10,11,12,13,14,15,16]. 

*Vespa velutina nigrithorax*, originating from Zhejiang Province in southern China, first invaded the Bongnae Mountain area (Busan, Korea) in 2003 through a trade ship. *Vespa velutina nigrithorax* is known to be a hunter and killer of bees. This species causes serious economic effects, including the loss of approximately 30% of bee colonies in apiaries. It can also sting people in urban areas, and is therefore a public health concern [17,18], and causes disturbances within the ecosystem [19,20,21]. Thus, it exerts very comprehensive invasive effects. Various studies are being conducted not only on methods to combat the alien species that has invaded Korean and European countries [22,23], but also on the use of these species as resources. Functionally, *Nidus vespae* (nests of social wasps) have been reported to exert pharmacological effects, such as anti-microbial, anti-viral, anti-inflammatory, and anti-tumor effects [24]. However, the anti-obesity effects of *Nidus vespae* extract (NV) have not yet been investigated. Therefore, in the present study, we aimed to evaluate the anti-obesity effects of NV and explore its underlying potential molecular mechanisms via in vivo and in vitro studies.

## 2. Methods and Materials

### 2.1. NV Preparation

NV was prepared using a nest of *Vespa velutina nigrithorax*, which was found on the campus of Kyungpook National University (Sangyeok-dong, Daegu, Korea) in mid-September 2018. The nest was located at a height of approximately 10 m, and there were approximately 900 adults in the nest. The nest was approached with a crane, the entrance was blocked, the branches connected to the nest were cut, and the nest was placed in a steel cage and brought to the laboratory in the School of Applied Biosciences, Kyungpook National University. After maintaining the cage in the deep freezer at −75 °C for 24 h, the nest was split to separate the individuals from the nest. Most of the nest materials used in the analysis included envelopes and some combs. They were maintained in a net for sufficient drying and were naturally dried at 25 °C for approximately one week (Figure 1).

*Nidus vespae* was crushed into <1 cm long pellets. The pellets (836 g) were extracted with a solvent mixture (15 L) composed of dichloromethane and MeOH (1:1, *v/v*) at room temperature (RT). The extract was filtered and concentrated under reduced pressure. The resultant crude extract (10.03 g) was suspended in MeOH (1 L), and the same volume of hexane was added. The sample was partitioned between the hexane (3.0 g) and MeOH layers using a separation funnel. The MeOH layer was concentrated under reduced pressure, and the obtained layer (6.27 g) was dissolved in distilled water (DW, 1 L). The DW suspension was extracted twice with ethyl acetate (EA, 1 L). Solvents were removed from the EA layer (1.66 g) under reduced pressure. The rest of the DW suspension was extracted twice with butanol (BuOH, 1 L). The solvent was evaporated under reduced pressure, and then the resultant extract was lyophilized to obtain powder (8.3 g). 

### 2.2. Reagents

Dulbecco’s Modified Eagle’s Medium (DMEM), fetal bovine serum (FBS), and newborn calf serum (NBCS) were purchased from Gibco Life Technologies (Grand Island, NY, USA). Insulin, indomethacin, dexamethasone (DEX), 3-isobutyl-1-methylxanthine (IBMX), 3-(4,5)-dimethylthiazol-2-yl-2,5-diphenyltetrazolium bromide (MTT), and Oil Red O (ORO) solution were purchased from Sigma-Aldrich (St. Louis, MO, USA). The 3T3-L1 preadipocytes were purchased from the Korea Cell Line Bank (KCLB, Seoul, Korea). For the Western blot analysis, the following antibodies were used: peroxisome proliferator-activated receptor gamma (PPARγ) and adiponectin (Abcam, Cambridge, MA, USA); CCAAT/enhancer binding protein alpha (C/EBPα), insulin receptor β (IRβ), and anti-phosphorylated IRβ (*p*-IRβ) (Cell Signaling Technology, Beverly, MA, USA); and protein kinase B (AKT), phospho-AKT (*p*-AKT), and β-actin (Santa Cruz Biotechnology, Dallas, TX, USA).

### 2.3. Animals and NV Administration

Six-week-old male C57BL/6 mice were purchased from Hyochang Science (Daegu, Korea). The mice were housed in cages for 1 week under a 12 h light/dark cycle at 25–30 °C for acclimatization. The mice were then randomly divided into the following four groups (*n* = 7 per group) according to diet: (1) normal diet (ND), (2) HFD, (3) HFD supplemented with NV at 100 mg/kg/day (referred to as NV 100), and (4) HFD supplemented with NV at 200 mg/kg/day (referred to as NV 200). A high-fat diet (HFD) with 60% fat (D12492; 5.2 kcal/g) was purchased from Research Diets Inc. (New Brunswick, NJ, USA). Throughout the study, NV was orally administered to the treated mice daily, whereas an equal volume of sterile water was administered to the control mice. Food and water were provided *ad libitum* for 8.5 weeks. Body weight and feed intake were measured every three days throughout the study. At the end of the experimental procedures, organs and blood were collected and utilized in subsequent analyses. This animal experiment was approved by the Institutional Animal Care Committee of Kyungpook National University, Daegu, South Korea (approval number: KNU 2019-0152). All methods were performed in accordance with the relevant guidelines and regulations.

### 2.4. Tissue Hematoxylin and Eosin (H&E) Staining

The histological analysis was performed as previously described [25]. Briefly, the inguinal white adipocyte tissue (iWAT) and livers were embedded in paraffin, sectioned (5 μm thickness), stained with H&E according to standard protocol, and subsequently examined under a microscope (Leica, Wetzlar, Germany).

### 2.5. Blood Biochemical Analysis

Plasma was prepared from whole blood via centrifugation. Aspartate aminotransferase (GOT), alanine aminotransferase (GPT), alkaline phosphatase (ALP), total bilirubin (T-BIL), blood urea nitrogen (BUN), creatinine (CREA), high-density lipoprotein (HDL), low-density lipoprotein (LDL), TGs, and cholesterol (CHO) in the plasma were measured using an Olympus AU400 analyzer (Olympus Optical, Tokyo, Japan), according to the manufacturer’s instructions.

### 2.6. Adipocyte Differentiation

Mouse 3T3-L1 preadipocytes were maintained in DMEM supplemented with 10% NBCS and 1% penicillin-streptomycin at 37 °C in a humidified 5% CO_2_ incubator. Before the experiment, 3T3-L1 preadipocytes were seeded and grown to confluence for 2 d. At day 2 post-confluence, the medium was changed to DMEM supplemented with an adipogenic cocktail—0.5 mM IBMX, 0.25 µM DEX, 167 nM insulin, 100 µM indomethacin, and 10% FBS—for 2 d in order to initiate adipocyte differentiation. Subsequently, stimulated cells were maintained in medium supplemented with 1.72 nM insulin and 10% FBS for an additional 6 d by replacing half the medium every 2 d. To examine the effects of NV, 3T3-L1 cells were treated with NV at 100 or 200 µg/mL every 2 d throughout the 8 d differentiation process.

### 2.7. ORO Staining and TG Assay

ORO staining and the TG assay were performed as previously described [26]. After 3T3-L1 preadipocytes were differentiated in the absence or presence of NV, the adipocytes were washed, fixed, and stained using ORO solution. The TG content was determined using a TG Quantification Kit (BioVision, Milpitas, CA, USA), according to the manufacturer’s instructions.

### 2.8. MTT Assay

3T3-L1 preadipocytes were seeded into 96-well plates and incubated at 37 °C overnight. The cells were then treated with either NV or an equivalent amount of DMSO for 48 h. MTT solution was added to each well, and the plates were incubated at 37 °C for 3 h. The reaction products were dissolved in isopropyl alcohol (Duksan Pure Chemicals, Ansan, Korea), and the absorbance of the reaction was measured at 595 nm.

### 2.9. Western Blot Analysis

Tissues and cells were homogenized with radioimmunoprecipitation assay lysis buffer containing phosphatase and protease inhibitors, and the lysates were centrifuged at 13,000 rpm for 15 min. Total protein (30 µg) was separated on 7.5–12% SDS-polyacrylamide gel, transferred onto nitrocellulose membranes, blocked with 5% non-fat skim milk, and incubated overnight with primary antibodies at 4 °C. After incubation, the membranes were washed with TBS-T buffer (10 mM tris, 150 mM NaCl, and 0.05% Tween 20) and were then incubated at RT with horseradish peroxidase-conjugated secondary antibody for 1 h. Protein signals were detected using an enhanced chemiluminescence kit (GE Healthcare, Amersham, Bucks, UK) with a Fusion Solo Detector (VilberLourmat, Marne-la-Vallée, France).

### 2.10. Real-Time Reverse Transcription Polymerase Chain Reaction (RT-PCR)

The total RNA from adipose tissue and 3T3-L1 adipocytes was isolated using an RNeasy Lipid Tissue Mini Kit (Qiagen, Hilden, Germany) and RNAiso Plus reagent (TaKaRa Bio, Japan), according to the manufacturers’ instructions. The total RNA (250 ng) was then used for complimentary DNA synthesis using the Prime-Script RT Reagent Kit (TaKaRa Bio, Japan). RT-PCR was performed on an iCycleriQ™ Real-Time PCR Detection System (Bio-Rad Laboratories, Hercules, CA, USA) using SYBR Green (TOYOBO, Japan). All primers were synthesized by Macrogen (Seoul, Korea). The primer sequences are shown in Table 1.

### 2.11. Statistical Analysis

Statistical data analysis was performed using SPSS 23 software (SPSS, Chicago, IL, USA). The differences between the means of the groups were analyzed by one-way analysis of variance. *p* values < 0.05 were considered statistically significant. 

## 3. Results

### 3.1. NV Suppressed Adipose Tissue Expansion and Reduced Plasma Lipid Levels in HFD-Induced Obese Mice

NV was prepared using a nest of *Vespa velutina nigrithorax*, as shown in Figure 1A,B. First, we examined the anti-obesity effects of NV in HFD-induced obese mice. Whole-body weight did not differ significantly between the NV and HFD groups, but there was a slight decrease in weight according to the concentration of NV administered (Figure 2A). Nevertheless, we observed that the NV 200 group showed reduced visceral and subcutaneous adipose tissues and higher feed intake compared to the HFD group (Figure 2B). 

These observations indicate that NV administration may affect HFD-induced metabolic abnormalities, such as fat accumulation and increased blood lipid levels. The NV 200 group showed remarkably reduced weights of iWAT and epididymal and retroperitoneal WATs (Figure 2C). In particular, both NV groups showed dramatically diminished iWAT weights compared to the HFD group. There were no significant differences in the weights of other organs, including the liver, heart, spleen, pancreas, lung, muscle, and kidney, between the NV and HFD groups (Figure 2D).

Next, we analyzed the effects of NV on plasma lipid profiles. The levels of LDL, TG, and CHO were significantly lower in the NV-treated mice compared to the HFD-fed mice. HDL levels increased in a dose-dependent manner in the NV groups compared with HDL levels in the HFD group (Figure 2E). Additionally, the levels of GOT and GPT (biochemical markers of liver dysfunction) decreased in a dose-dependent manner in the NV groups compared with those in the HFD group (Table 2). Meanwhile, no significant differences in nephrotoxicity marker levels (ALP, T-BIL, BUN, and CREA) were observed between the NV and HFD groups.

Collectively, these results suggest that NV demonstrated anti-obesity effects, especially the regulation of abnormal adipose tissue expansion and lipid content, in HFD-induced obese mice.

### 3.2. NV Induced Adipose Browning and Ameliorated Liver Steatosis

Next, we investigated whether NV administration affected adipocyte hypertrophy and hepatic steatosis, both of which are common metabolic complications associated with obesity. Both adipocyte size and liver lipid deposition were lower in the NV groups compared to the HFD group (Figure 3A). The NV and ND groups showed similar dose-dependent patterns of adipocyte size distribution (Figure 3B).

Adipocyte size reflects adipocyte type, and decreasing size is associated with adipocyte browning [27]. Similar to brown adipocytes, beige adipocytes in WAT are characterized by the presence of multilocular lipid droplets [28]. Considering the histological features of adipocytes in the NV-treated mice, we hypothesized that NV might induce adipose browning via thermogenic gene overexpression. As expected, the NV groups showed a dose-dependent increase in UCP1, PGC1α, and PRDM16 expression in iWAT (Figure 3C). On the other hand, the mRNA expressions of these genes did not change in BAT in the NV and control groups (Figure 3D). These results imply that NV administration affected lipid accumulation in metabolic organs and adipose browning specifically in WAT.

### 3.3. NV Inhibited Adipogenesis in 3T3-L1 Adipocytes without Inducing Cytotoxicity

Considering the demonstrated anti-obesity effects of NV in the adipose tissues of HFD-induced obese mice, we speculated that NV treatment could control adipocyte differentiation in vitro. NV significantly inhibited adipocyte differentiation and intracellular TG accumulation in a dose-dependent manner (Figure 4A–C). To confirm whether the anti-differentiation effects of NV were caused by cell death or inhibited proliferation, we examined the cytotoxicity of NV in preadipocytes. Treatment with NV at 100 and 200 μg/mL for 48 h did not affect preadipocyte viability (Figure 4D). These results indicated that NV might exert potent anti-differentiation effects on 3T3-L1 adipocytes without inducing cytotoxicity to preadipocytes.

### 3.4. NV Suppressed the Activation of Insulin Signaling and the Expression of Adipogenesis-Related Proteins

Finally, we investigated the mechanisms underlying the anti-differentiation effects of NV. Insulin is critical in the mediation of adipogenesis since it triggers the activation of IR and AKT [29,30]. Consequently, AKT is upstream of key adipogenesis regulatory genes, such as PPARγ, C/EBPα, and adiponectin [31]. Compared with no treatment, NV treatment significantly suppressed *p*-IRβ and *p*-AKT expression in a dose-dependent manner (Figure 5A–C). Furthermore, NV treatment reduced PPARγ, C/EBPα, and adiponectin mRNA expression (Figure 5D). Consistent with mRNA expression, protein expression decreased in NV-treated adipocytes (Figure 5E). These results implied that NV treatment inhibited adipocyte differentiation by regulating the insulin signaling cascade.

## 4. Discussion

Several studies have demonstrated that a variety of natural products, including natural extracts and compounds isolated from plants, can increase weight loss and prevent diet-induced obesity [32]. Recent research suggests that insects and other neglected resources display various biological activities, providing critical information for the development of new drugs for the prevention and treatment of metabolic diseases [33,34]. *Nidus vespae* is the nest of *Polistes olivaceus* and is used in Chinese folk medicine to treat a variety of diseases, including cardiovascular, digestive, and urinary disorders [35].

In the current study, we demonstrated the anti-obesity effects of NV in in vivo and in vitro systems using an HFD-induced obese mouse model and 3T3-L1 adipocytes, respectively. Although NV administration did not affect whole-body weight, we showed that it significantly reduced adipose tissue mass. Mice treated with 200 µg/mL of NV showed significantly increased food intake, implying that NV possibly regulated energy expenditure and metabolism, and exerted protective effects against abnormal adipose tissue expansion in HFD-induced obese mice. Beige adipocyte plasticity is important for feeding-associated changes in energy expenditure [36]. In the current study, we found that UCP1 expression was higher in the iWAT of NV-treated mice than in that of HFD-fed mice. This finding suggests that NV administration may accelerate energy expenditure by activating UCP1-positive beige adipocytes.

Previous studies have reported that several natural products suppress insulin-stimulated adipogenesis in adipocytes [37,38]. The mechanisms mediating the effects of NV revealed that IR and AKT activation was strongly decreased and downstream target genes, including PPARγ, C/EBPα, and adiponectin, were downregulated in 3T3-L1 adipocytes. Altogether, these results indicated that NV treatment inhibited adipose expansion and adipocyte differentiation, and enhanced adipose browning, which ultimately lead to the suppression of obesity.

## 5. Conclusions

Our findings demonstrated that NV administration exerted anti-obesity effects in HFD-induced obese mice and 3T3-L1 adipocytes. These effects included the benefits of adipose hypertrophy and browning, hepatic steatosis, adipocyte differentiation, and the regulation of adipogenesis-related gene expression. To our knowledge, the present study provides the first evidence that NV exerts anti-obesity effects potentially by regulating the insulin signaling cascade.

## Figures and Tables

**Figure 1 biology-11-01013-f001:**
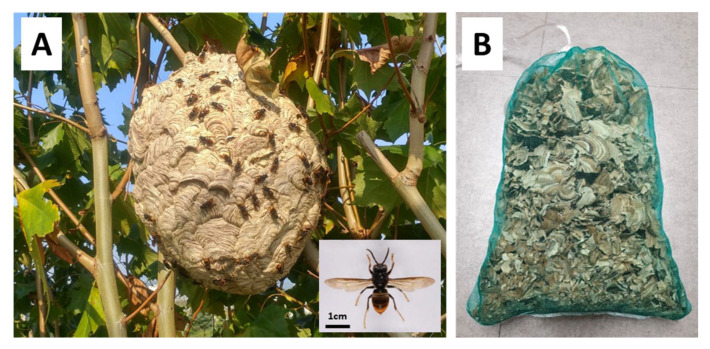
The *Vespa velutina nigrithorax* nest (**A**) and *Nidus vespae* extract materials (**B**).

**Figure 2 biology-11-01013-f002:**
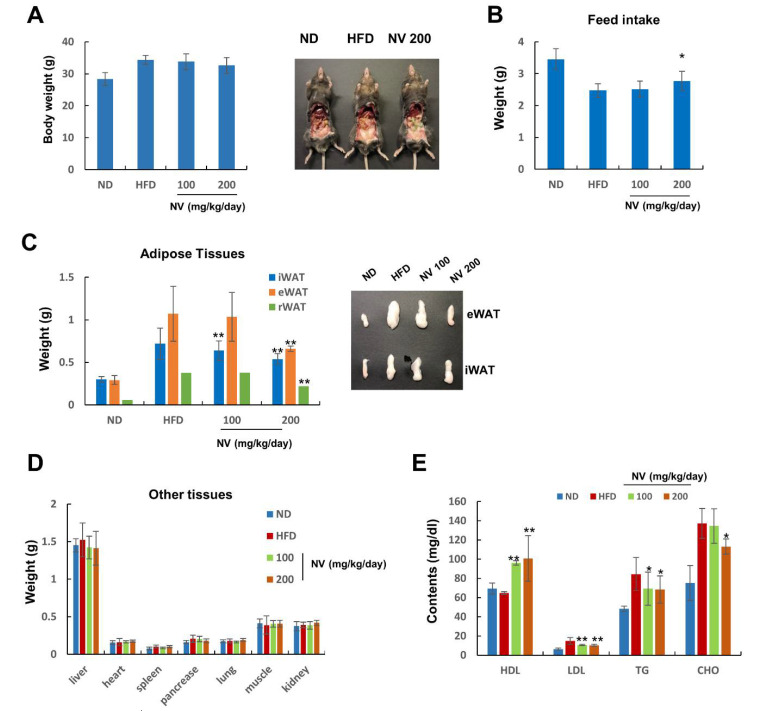
Effects of NV on adipose tissue expansion and plasma lipid content in HFD-induced obese mice. The ND group (n = 7) was fed a normal diet, the HFD group (n = 7) was fed an HFD, and the NV 100 and NV 200 groups (n = 7) were fed an HFD plus NV at the indicated concentrations (100 and 200 µg/mL, respectively). (**A**) Body weight was measured at the end of the animal study (left). Representative images of the abdominal organs from the indicated mice groups are shown (right). (**B**) Food intake was recorded every 3 days throughout the study. (**C**) Adipose tissue weight (left) and representative eWAT and iWAT images (right) are shown. (**D**) Liver, heart, spleen, pancreas, lung, muscle, and kidney weights are shown. (**E**) Plasma HDL, LDL, TG, and CHO levels were measured. Significant differences from the HFD group are indicated by ** *p* < 0.01 and * *p* < 0.05. Bar graphs show the mean ± standard deviation from five individual mice per group. NV, *Nidus vespae* extract; HFD, high-fat diet; eWAT, epididymal white adipose tissue; iWAT, inguinal white adipose tissue; rWAT, retroperitoneal white adipose tissue; HDL, high-density lipoprotein; LDL, low-density lipoprotein; TG, triglyceride; CHO, cholesterol.

**Figure 3 biology-11-01013-f003:**
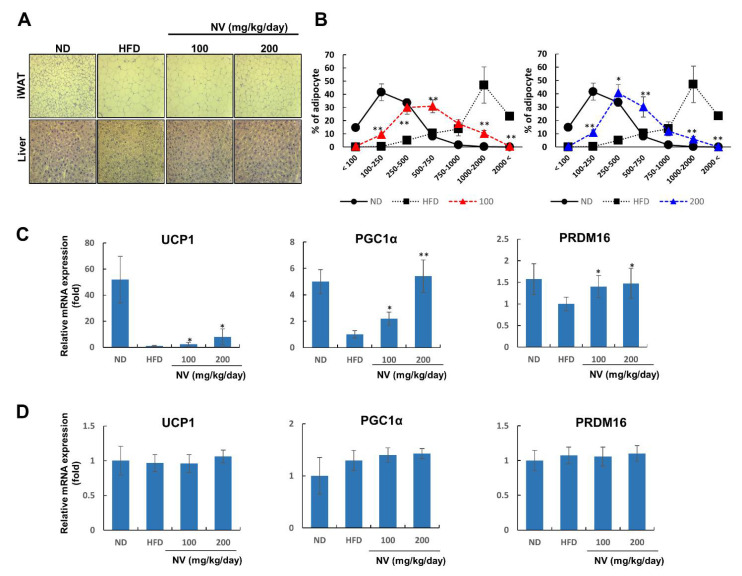
NV reduced the size of adipocytes in adipose tissue and suppressed lipid accumulation in the liver. The ND group (n = 7) was fed a normal diet, the HFD group (n = 7) was fed an HFD, and the NV 100 and NV 200 groups (n = 7) were fed an HFD plus NV at the indicated concentrations (100 and 200 µg/mL, respectively). (**A**) Representative images of iWAT (upper) and liver tissues (lower) stained with hematoxylin and eosin were taken at a magnification of 20×. (**B**) Adipocyte size was measured using ImageJ software. Density curves of ND, HFD, and NV 100 groups (left) or the NV 200 group (right) are shown. (**C**,**D**) UCP1, PGC1α, and PRDM16 mRNA expression in iWAT (**C**) and BAT (**D**). Significant differences from the HFD group are indicated by ** *p* < 0.01 and * *p* < 0.05. Bar graphs show the mean ± standard deviation from five individual mice per group. NV, *Nidus vespae* extract; HFD, high-fat diet; iWAT, inguinal white adipose tissue; UCP1, uncoupling protein 1.

**Figure 4 biology-11-01013-f004:**
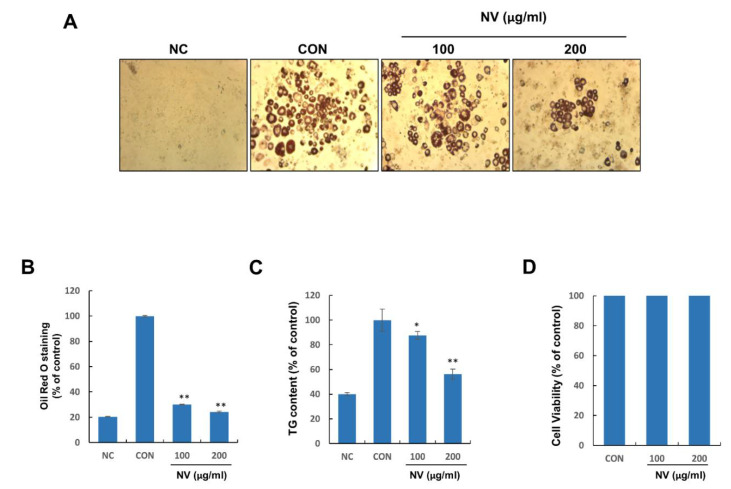
NV inhibits the differentiation and intracellular lipid accumulation of 3T3-L1 adipocytes without inducing cytotoxicity to preadipocytes. 3T3-L1 preadipocytes were treated with differentiation medium in the presence or absence of NV at 100 or 200 μg/mL for 8 days. (**A**) Differentiated adipocytes were photographed at a magnification of 40×. (**B**) The Oil Red O solution content was measured at an absorbance of 450 nm. (**C**) The intracellular TG content was assessed at an absorbance of 570 nm. (**D**) 3T3-L1 preadipocytes were treated with NV at the indicated concentrations, and cell viability was determined by the MTT assay using a spectrophotometer at a wavelength of 595 nm. Preadipocytes were used as the negative control (NC) and fully differentiated adipocytes were used as the positive control (CON). The NC and CON were treated with the same amount of DMSO instead of NV. All cell culture experiments are representative of three independent experiments. Data are presented as the mean ± standard error of the mean. Significant differences from the CON group are indicated by ** *p* < 0.01 and * *p* < 0.05. NV, *Nidus vespae* extract; TG, triglyceride.

**Figure 5 biology-11-01013-f005:**
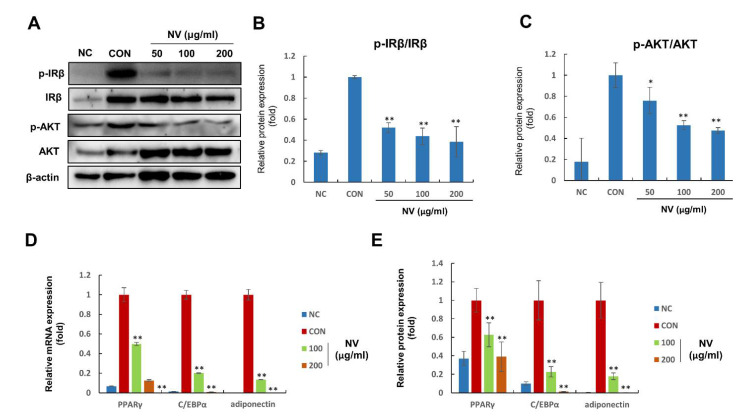
NV modulates the mRNA and protein expression of adipogenesis-related genes in 3T3-L1 adipocytes. 3T3-L1 adipocytes were treated with NV at the indicated concentrations (50, 100, and 200 µg/mL) for 8 days, and total RNA and protein were isolated. Preadipocytes were used as the negative control (NC) and fully differentiated adipocytes were used as the positive control (CON). The NC and CON were treated with the same amount of DMSO instead of NV. (**A**–**C**) *p*-IRβ, IRβ, *p*-AKT, AKT, and β-actin protein expression is shown (**A**). The relative expression was quantified using Fusion Solo Detector software and was expressed as ratios of *p*-IRβ/IRβ (**B**) and *p*-AKT/AKT (**C**). (**D**,**E**) PPARγ, C/EBPα, and adiponectin mRNA (**D**) and protein (**E**) expression is shown. Each experiment was repeated in triplicate. Data are presented as the mean ± standard error of the mean. Significant differences from the CON group are indicated by ** *p* < 0.01 and * *p* < 0.05. NV, *Nidus vespae* extract; IRβ, insulin receptor β; *p*-IRβ, anti-phosphorylated IRβ; AKT, protein kinase B; *p*-AKT, phospho-AKT; PPARγ, peroxisome proliferator-activated receptor gamma; C/EBPα, CCAAT/enhancer binding protein alpha.

**Table 1 biology-11-01013-t001:** Primer sequences used for quantitative RT-PCR.

Gene	Forward Primer	Reverse Primer
PPARγ	GTAATCAGCAACCATTGGGTCA	GAAAGGTTGGCTTGACCTGCT
C/EBPα	GGTCGTTTCTCCATTAAATTCTCAT	CTAGAAACTTTCCCAGAAATCTTCC
adiponectin	GATGGCACTCCTGGAGAGAA	GCGAAACTCGATGACTCCTCGG
β-actin	CGTGCGTGACATCAAAGAGAA	GCTCGTTGCCAATAGTGATGA.

**Table 2 biology-11-01013-t002:** Plasma biochemical values. Significant differences from the HFD group are indicated by ** *p* < 0.01 and * *p* < 0.05.

	ND	HFD	NV 100	NV 200
GOT (U/L)	62.4 ± 16.79	163.5 ± 9.27	119.4 ± 20.87 *	86.9 ± 34.05 **
GPT (U/L)	21 ± 1.73	48 ± 8.14	32.7 ± 1.15 **	27.0 ± 1.0 **
ALP (U/L)	96 ± 1.15	73 ± 2.0	66.8 ± 2.30	66.7 ± 18.29
T-BIL (mg/dL)	0.1 ± 0.08	0.1 ± 0.0	0.1 ± 0.0	0.2 ± 0.12
BUN (mg/dL)	21.8 ± 3.59	22.4 ± 0.4	23.8 ± 2.65	21.9 ± 2.38
CREA (mg/dL)	0.6 ± 0.08	0.6 ± 0.0	0.6 ± 0.0	0.5 ± 0.09

## Data Availability

The data presented in this study are available on request from the corresponding author.

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
