# Peer review of "Nidus vespae Built by an Invasive Alien Hornet, Vespa velutina nigrithorax, Inhibits Adipose Tissue Expansion in High-Fat Diet-Induced Obese Mice"

_biology, 2022, doi:10.3390/biology11071013_

Round 1

Reviewer 1 Report

This study focused on investigating the potential anti-obesity effects of Nidus vespae. The topic was somewhat interesting. However, due to lack of details highly affect the rationality of the evidence of the anti-obesity effects of NV.

First of all, I suggest that it is better to provide much more specific details, for example, What is the high-fat diet (HFD)? What percentage of calories is in HFD? How long were these mice fed HFD? How many mice are in each group? How often do the mice be weighed?

Since the significance is relating to browning of WAT in mice, the author should describe the details on brown adipose tissue itself. Furthermore, the author needed to explore evidence in energy expenditure depending on different ambient temperatures.

Author Response

Dear reviewer

The authors of this paper appreciate the intense review of this work. Comments, suggestions, and recommendations. We have studied comments carefully and have made correction which we hope meet with approval. The main corrections in the paper and the responds to the reviewer’s comments are as flowing:

  1. This study focused on investigating the potential anti-obesity effects of Nidus vespae. The topic was somewhat interesting. However, due to lack of details highly affect the rationality of the evidence of the anti-obesity effects of NV.

RESPONSE: Thank you very much for your comment. We modified the introduction section to emphasize research background and scientific rationality.

  1. First of all, I suggest that it is better to provide much more specific details, for example, What is the high-fat diet (HFD)? What percentage of calories is in HFD? How long were these mice fed HFD? How many mice are in each group? How often do the mice be weighed?

RESPONSE: Thank you for your suggestion. We have to add the detailed experimental conditions and information in M&M section.

  1. Since the significance is relating to browning of WAT in mice, the author should describe the details on brown adipose tissue itself. Furthermore, the author needed to explore evidence in energy expenditure depending on different ambient temperatures.

RESPONSE: Thank you for your suggestion. According to your suggestion, we analyzed the expression of browning-related genes such as UCP-1, PGC1a, and PRDM16 in BAT (Brown adipose tissues) and observed that significant differences were not observed in the expression of these genes between NV and control groups. This data imply that NV did not affect BAT defect and brown adipocytes function. We added this result as shown in Fig. 3D.

Reviewer 2 Report

The title of this study is: Nidus vespae built by an invasive alien hornet, Vespa velutina nigrithorax, inhibits adipose tissue expansion in high-fat dietinduced obese mice. In this study aimed to evaluate the anti-obesity effects of NV and explore the underlying potential molecular mechanisms via in vivo and in vitro studies.

I commented on the manuscript and the comments are presented below:

Part 1: Introduction.

The Introduction to the study is broad and does end with a clearly stated purpose or goals that the Authors wish to pursue.

Part 2: Methods and Material

The Methods section provides the reader with enough information to repeat the experiments conducted. Only the basic statistical analysis was used to describe the differences. Have the Authors attempted to use other more comprehensive statistical analyzes, e.g. principal components analysis of PCA? With such a large number of parameters tested, which may affect the characteristics examined, the Principal Component Analysis (PCA) should be used to results analyzed. More advanced statistical analysis should be performed. The use of advanced statistical methods to fully describe the relationship between the parameters studied and the aspects of the research work carried out in the presented manuscript. The Authors can determine the strength of the influence of a particular parameter on the variance of the system. At the same time, correlation relationships between the determined parameters can be determined.

Part: 3 Results

For the most part the Results section is well structured.

Part: 4 Discussion

In the Discussion chapter, there is full comparison and confrontation with the research of other authors in this area. The Conclusions chapter contains information obtained after conducting experiments but performing only base statistical analyzes.

Part: References.

The literature used is appropriate, but it can be supplemented with items from recent years of publications on a similar problem.

Author Response

Dear reviewer

The authors of this paper appreciate the intense review of this work. Comments, suggestions, and recommendations. We have studied comments carefully and have made correction which we hope meet with approval. The main corrections in the paper and the responds to the reviewer’s comments are as flowing:

Part 1: Introduction.

The Introduction to the study is broad and does end with a clearly stated purpose or goals that the Authors wish to pursue.

RESPONSE1 : Thank you very much for your comment. We modified the introduction section to express the research background and a clear statement of the problem.

Part 2: Methods and Material

The Methods section provides the reader with enough information to repeat the experiments conducted. Only the basic statistical analysis was used to describe the differences. Have the Authors attempted to use other more comprehensive statistical analyzes, e.g. principal components analysis of PCA? With such a large number of parameters tested, which may affect the characteristics examined, the Principal Component Analysis (PCA) should be used to results analyzed. More advanced statistical analysis should be performed. The use of advanced statistical methods to fully describe the relationship between the parameters studied and the aspects of the research work carried out in the presented manuscript. The Authors can determine the strength of the influence of a particular parameter on the variance of the system. At the same time, correlation relationships between the determined parameters can be determined.

RESPONSE2 : Thank you for your suggestion. In the present study, we showed the results for anti-obesity effects of NV using animal model and cell line. In these results, we separated the experimental group from at least 2 to maximal 5 groups and repeated the experiments in triplicate. However, the results did not contain any meta-size and omics data. Therefore, we used one-way ANOVA using SPSS software to determine the statistically significant of all results presented.

Part: 3 Results. For the most part the Results section is well structured.

RESPONSE3 : Thank you very much for your review.

Part: 4 Discussion

In the Discussion chapter, there is full comparison and confrontation with the research of other authors in this area. The Conclusions chapter contains information obtained after conducting experiments but performing only base statistical analyzes.

RESPONSE4 : Thank you for your review and suggestion. As described also RESPONSE2, our results did not contain any omics and correlation coefficient data. Therefore, we have to use only base statistical analyzes. We will be appreciated if reviewer understands it’s too difficult to express our data using other more comprehensive statistical analyzes.

Part: References.

The literature used is appropriate, but it can be supplemented with items from recent years of publications on a similar problem.

RESPONSE5 : Thank you for your comment. We added some literature published in recent years.

Round 2

Reviewer 2 Report

The authors referred to the comments from the previous review for the manuscript titled: Nidus vespae built by an invasive alien hornet, Vespa velutina nigrithorax, inhibits adipose tissue expansion in high-fat diet-induced obese mice. I accept explanations. In the future, I suggest using more precise  describing relationships between the parameters studied. They supplemented the discussion with a new literature data strengthens the message and importance of information in the manuscript.